# Effects of a Self-Regulated Training Program on the Repeated Power in Female College Handball Players: An Intervention Study

**DOI:** 10.3390/ijerph182312662

**Published:** 2021-12-01

**Authors:** Sebastián Espoz-Lazo, Claudio Hinojosa-Torres, Claudio Farías-Valenzuela, Sebastián Álvarez-Arangua, Paloma Ferrero-Hernández, Pedro Valdivia-Moral

**Affiliations:** 1Facultad de Ciencias Para el Cuidado de la Salud, Universidad San Sebastián, Lota 2465, Providencia 7510157, Chile; sebastian.espoz@uss.cl; 2Facultad de Ciencias de la Actividad Física y del Deporte, Universidad de Playa Ancha, Valparaíso 2360072, Chile; 3Instituto del Deporte, Universidad de Las Américas, Santiago 9170022, Chile; cfaria46@edu.udla.cl; 4Departamento de Didáctica de la Expresión Músical, Plástica y Corporal, Universidad de Granada, 18010 Granada, Spain; pvaldivia@ugr.es; 5Exercise and Rehabilitation Sciences Laboratory, School of Physical Therapy, Faculty of Rehabilitation Sciences, Universidad Andres Bello, Santiago 7591538, Chile; s.alvarezarangua@uandresbello.edu; 6Facultad de Educación y Cultura, Universidad SEK, Santiago 7520318, Chile; paloma.ferrero@zonavirtual.uisek.cl

**Keywords:** RSSJA test, OMNI-RES, jumping, running, self-perceived effort, strength, resistance

## Abstract

Several resistance training programs using conventional methodologies have been implemented with the purpose of improving the ability to perform power actions in handball, especially during the competitive season. In contrast, methodologies based on a contemporary perspective, which considers the human being as a self-regulating biological entity, and designed specifically for female college players, are scarce. The aim of this research was to investigate the effects of an eight-week resistance training program, in which the athletes were able to control the loads according to their self-perceived effort and rest on their repeated shuttle sprint and jump ability. The sample was composed of 16 female players of a handball team from the faculty of physics and mathematics sciences of a Chilean university. The RSSJA test was used to evaluate players’ conditions pre- and post-training program, and the self-perceived effort scale called OMNI-RES was used for the prediction and control of loads. Results indicated that, after the application of an eight-week resistance training program, significant improvements *p* ≤ 0.05 on the jump height (pre: 1836.4 W; average post: 2088.9 W) and running speed (average pre: 3.2 m/s; average post: 4.0 m/s) were obtained, as well as a significant reduction in the loss of power and speed between each set of the applied test.

## 1. Introduction

Handball has traditionally been characterized by the need for repetitive actions such as throwing, blocking, pushing, running, changing of direction, and jumping [1], which are mainly dependent on the systematic ability to produce muscle power at high speed during the game [2]. Likewise, the movements of handball demand a high level of intensity which is maintained from the beginning to the end of each match, associated with its offensive technical–tactical actions and a large number of strong defensive contacts [3,4]. To satisfy these intensity and power needs, resistance training programs were implemented during the pre-competitive and competitive season. These have mainly been reported in the context of professional and elite sport and based on progressive plans, as well as prescribed in accordance with quantitative, objective, and standardized tests such the maximal repetition of strength [5,6,7,8]. 

On the other hand, in the last five years modern handball has begun to be described as a sport which develops through the logic of complex dynamic systems, in which interactions between players are essential for success and where all of the different components of each player involved are decisive for optimal individual and collective performance [9,10,11]. This perspective, driven by the philosophy of Seirul-Lo [12], understands the human being as a dynamic and multidimensional biological system, which unfolds and develops in continuous interaction with its constituent elements: physical, cognitive, coordinative, and emotional [13], as well as with the environment and other humans around it [14]. In this sense, new patterns of behavior are constantly emerging, which are affected by multiple external variables; these unbalance the homeostasis of this individual biological system and bring about new states of equilibrium [15]. 

Bearing in mind this contemporary perspective, strength training that considers the systemic characteristics of the human being in team sports is more recent and therefore less prevalent in published literature compared to training methodologies based on the traditional view. Nonetheless, important findings such as those of Tous-Fajardo et al. [16], based on this new perspective, explain that the strength training of soccer should be approached from a three-dimensional viewpoint by adding vectors for strength training to the movement (vertical and horizontal axes), displacement in rotary axis and acceleration and decelerations, which are executed at high intensities but limited to the athlete’s characteristics according to their own internal homeostatic regulations. In this sense, methodologies based on traditional strength training would not be as effective as training with isoinertial eccentric overload devices since this act according to the force applied by the subject and not by that determined by the coach. On the other hand, Gonzalo-Skok et al. [17] show how a strength training program based on the understanding of the human being as a self-regulating biological entity produces better results since it is the individual themself who determines the component of the load volume in the prescribed exercise. From the same perspective, the recent study by Arede et al. [18] shows how a group of basketball players increased their ability to generate power by availing of resting periods of time between each series chosen in the moment by the players themselves (each one rested the time they needed) compared to those who rested in a prescribed way (three minutes between each series). The authors explain these findings through the viewpoint of humans as autopoietic beings, with processes of self-regulation and self-regeneration that act at different speeds and in unique ways for each individual [19]. 

The studies above provide evidence for the effectiveness of this alternative means of designing and performing strength training in team sports. However, methodological options for strength training in female college handball players based on the arguments presented above have not been found. A review of the literature published in the last 10 years found that some studies focus simply on demonstrating effects on throwing power and maximum running speed [20,21]. Studies concerning general strength training are mainly focused on injury prevention [22,23]. Finally, articles that show results for repeated shuttle sprint and jump ability in female college handball players have been difficult to find. This research aims to address this gap in the literature through an intervention study investigating the effects of a resistance training program in which loads are controlled by the self-perceived efforts of a group of female Chilean university handball players on their repeated shuttle sprint and jump ability. In addition, the research seeks to determine whether this methodological approach, focused on self-perceived efforts to control strength training loads in female handball players, could offer a useful approach for investigating performance in other non-professional contexts. 

## 2. Materials and Methods

### 2.1. Design and Participants

This is an intervention study drawing from a specific population [24] of handball players belonging to the same university-level team who were also students of the faculty of physical and mathematical sciences of a Chilean university. As an inclusion criterion, an experience of at least one semester in sport was requested. Those players who were in the post-injury rehabilitation process were excluded since they held sessions of physical activity parallel to regular training, which, in turn, were specialized for their recovery process. Thus, 16 players were selected of age: 20.25 + 2.2 years, height: 162.3 + 4.9 cm, mass: 62.0 + 8.4 kg., fat mass: 35.8 + 5.4%, muscle mass: 26.5 + 2.2%, and BMI: 23.6 + 2.3 kg/m^2^. From the group of selected players, there were two center-backs, three full-backs, six wingers, three goalkeepers, and two pivots.

### 2.2. Procedure

An initial evaluation was carried out, in which the anthropometric measurements of weight, height, % of fat mass, and lean mass were made through a standing stadiometer brand SECA^®^, model 206 and a Tanita brand OMNRON^®^ model Hbf514. In the same context, each player underwent the RSSJA test [25] to assess their ability to perform repetitive jumps and runs. At the end of the session, all the data were recorded in an Excel^®^ spreadsheet. For all of the above, all of the players appeared on the day of the initial evaluation, already having had previous experience in the applied test, having been evaluated at the end of the previous pre-season. They were requested not to eat food for an hour before the evaluation, and they were also required to maintain the same type of diet that they follow on a daily basis, without making modifications of any kind. Similarly, they were required not to ingest stimulant drugs or medicines such as caffeine or guarana before the evaluation. The players were permitted to hydrate ad-libitum. After the initial evaluation, the players were subjected to a power training program using, for load control, the OMNI-RES effort perception scale [26]. Once the training process was completed, the players were again evaluated under the same initial conditions described above. Before starting the procedures, all the athletes were duly informed of the objective and the protocol of the investigation. Likewise, they signed an informed consent where they declared their willingness to participate in the experiment and gave authorization for the data obtained to be published. 

At the time of participating in the study, the players were in the initial phase of the competitive season in the so-called group phase of the university’s inter-university tournament, competing at a rate of one match per week (only on Saturdays), with a training volume of three weekly handball sessions each lasting an hour and a half, where the technical–tactical and strategic fundamentals of the competition were worked on, without any other hour of physical activity per week.

### 2.3. OMNI-RES Scale Training Program

After the initial evaluation, the players underwent eight weeks of strength training, particularly focused on the development of power, using the OMNI-RES effort perception scale for prediction and control of loads [26], each player training independently and individually, without the company of a physical trainer. For this, an induction session was carried out during which they were given the training plan and were taught to correctly execute the exercises and how to use the load control instrument. The indications regarding the execution of the strength training plan were that, in each execution of the respective repetitions, the action should be explosive and that the displaced mass loaded in the weights should generate an effort between six and seven according to the scale of OMNI-RES.

Similarly, the players were instructed to perform the number of series within the range described in the planning. To regulate this, they were told that they should feel fatigued when reaching the maximum number of repetitions and that it could not be so high that they did not reach the minimum number of repetitions, for which, they were told that they should modify the mass to be displaced by increasing it or decreasing it to maintain levels six to seven of the OMNI-RES scale by executing between four to six repetitions. For the rest between series, they were required to respect it in the same way as the time ranges between the minimum and the maximum to start executing the exercises when they felt ready to perform them successfully.

The strength training plan was structured around exercises associated with the fundamental movements of handball: throwing, defensive contact, runs, jumps, and change of direction. In the same way, they were organized so that each exercise was executed, working on different areas of the body alternately and not consecutively: upper body, lower body, and core. Each training session included power-stimulation exercises as well as compensatory exercises for the antagonist muscles. Finally, each microcycle had a duration of two weeks made up of a total of six training sessions carried out on non-consecutive days: Monday, Wednesday, and Friday (Table 1).

### 2.4. Relationship between Muscle Power and Repeated Running and Jumping

In the present study, the definition for muscle power is understood as the muscle’s ability to produce force in the lesser time possible due to a contraction. The higher the muscle’s ability to produce power, the faster the action of running and the higher the action of jumping [27]. However, as handball is an intermittent sport that requires maintaining repetitive actions during long periods of time, repeated running and jumping, and the ability to resist fatigue induced by these two frequent actions are more important to assess compared to evaluating the absolute level of force generated per se in each sprint and jump, because this does not represent the relationship between these actions and the fatigue that they produce [1]. Therefore, because repeated sprints and jumps are less dependent on anaerobic metabolism compared with only one bout of a sprint and jump [28], the variables to assess are focused on what it is functional to handball, which is the repeated shuttle sprint and jump ability. 

### 2.5. Repeated Shuttle Sprint and Jump Ability Test

The test applied to evaluate the ability to perform jumps and sprints in a repetitive way was according to the described protocol [25], which contemplates the execution of six series of two sprints at maximum speed (SMS) traveling 12.5 m, each followed by a deceleration that ends with a countermovement jump (CMJ) on a jumping platform and with an active recovery jogging, managing to cover a total distance of 36 m. The entire series must be performed within 25 s to ensure an average speed of 2.1 m/s to then start the next repetition until the end of the protocol.

Photocells, the jump platform, and the ChronoJump Bosco System software, all manufactured by Chronojump® in Barcelona, Spain, were used for the control of the SMS, as well as for the measurement of the CMJ. Particularly, for the execution of the CMJ, players were requested to keep their hands constantly on their hips and the depth of the squat was self-selected as there was no influence of this on the height of the jump [29]. The data obtained regarding the running time of the sprints and the height of jumps were recorded in an Excel^®^ data sheet.

### 2.6. Analysis of Data

For data analysis, the IBM SPSS Statistic Software^®^ v25.0 program (SPSS Inc., Chicago, IL, USA) was used, with which the results of SMS execution time (SMStime) and the height of the CMJ (CMJwatts) were obtained in the pre- and post-test evaluations were compared. Descriptive statistics were presented in mean and standard deviation (SMS mean ± SD and CMJ mean ± SD) with the normal distribution of the results according to the Shapiro–Wilk test. Subsequently, the T Student test was applied for related samples with the objective of comparing the pre- and post-intragroup means of CMJ watts and SMS time for each series executed.

## 3. Results

The data obtained indicate that for each series of the RSSJA test, particularly in the SMSs, a significant decrease in the duration times was achieved in each 25 m run. Similarly, for the CMJ, a significant improvement in the power exerted for each of the jumps was observed when comparing the means of the pre- and post-training program evaluations. Along with this, it is possible to see that prior to the training program, there was a loss of speed between each of the sprints, reaching a difference of +0.97 s between the first and the last. This result was reduced in the evaluation after the training program, where the loss of speed between sprints was less, obtaining a difference of +0.68 s between the first and the last sprint. The same phenomenon occurred with respect to the CMJ jumps, where a loss of power was obtained between each of the series, reaching a difference of −153.91 W when comparing the first with the last jump in the pre-evaluation. In the post-evaluation, the reduction in power between each jump decreased, obtaining a loss of −89.15 W between the first and the last jump (Table 2).

As the *p*-value does not represent the outcome of this intervention by itself, the effect size was also measured with Cohen’s d. Results indicate that for each player, a large effect was achieved for both the CMJ and the SMS, except for three players in the sprint at maximal speed, where two achieved a medium effect and one a small effect correspondingly (Table 3). 

After the intervention, it was found that all players had improved in both CMJ and SMS, although at different rates. However, as shown, all improvements were significant (Figure 1 and Figure 2). 

## 4. Discussion

The objective of this research was to describe the effects of a strength training program in which loads were controlled by the self-perception of female college handball players on their repetitive capacity to generate power. It also aimed to determine whether research focused on self-perceived efforts to control strength training loads in female handball players could be valuable as a means of investigating different non-professional contexts. 

The main finding of this study was that the program generated an increase in the jump power and the speed of the sprints, as well as a decrease in the loss of power when performing the actions iteratively. This offers a competitive advantage that increases the chances of success, since the systematic and repetitive execution of power actions is one of the fundamental characteristics for victory in handball competitions [1].

Looking in further detail at the results, a significant decrease in the times obtained in each of the CMV series executed during the RSSJA test could be observed. These results are rare since different magnitudes are usually reported in adaptive responses to standardized training programs [30]. For example, the study by Braz et al. (2018) [31] found that adult basketball players, who were trained under a program of resistant strength, maximum strength, and power for 12 weeks did not show significant improvements in the ability to perform SMS repetitively. However, in this study, when doing the analysis individually, it was shown that all players had presented improvements, some with greater effects than others, demonstrating the variability of the adaptive responses to a structured program based on standardized loads according to a previous evaluation of maximum force. In contrast, in this study, the development of the training program was controlled by the OMNI-RES scale, which enables the continual adaptation of the program loads to the individual responses of each player in order to maintain the desired effect, which would explain the improvement in the total of the SMS. 

On the other hand, regarding the decrease in the loss of speed of the SMS and the power of the CMJ between each of the series of the RSSJA test, in this study, the protocol of executing CMJ jumps during the recovery time in a series of repetitive SMS does not negatively affect performance and, in fact, allows effective recovery for the execution of the following series [32]. It is possible that the minor loss of speed and power between each of the series is due to an improvement in the recovery capacity, which may have been generated as a result of the designed training program. However, the phenomenon could also be explained by an improvement resulting from technical–tactical training, as handball is a sport with intermittent intensities. Forbes et al. (2008) [30] have shown that high-intensity training for short periods of time significantly improves the recovery of phosphocreatine, a fundamental substrate for SMS and CMJ. Both the training program of this research as well as the technical–tactical training feature the characteristics of being high intensity for short periods of time, executed repeatedly, such that it is complex to determine exactly what it was that specifically produced the adaptation that allowed the reduction of the loss of speed and power. However, the results of the study by Hammami et al. (2018) [33], in which they subjected two groups of soccer players to a training program, where the control group maintained their regular soccer training, while the experimental group performed an additional strength program for the lower body between 70 and 90% of the maximum strength evaluated, could explain the phenomenon obtained in this research. Hammami et al. found that none of the groups improved in the repetitive ability to execute SMS and CMJ, but that there were improvements in the experimental group in terms of jump power and speed, while the control group had no improvements in any of the groups. Bearing these results in mind, the possibility that our program is effectively responsible for the reduction in the loss of speed in the SMS and in the power of the CMJ is greater.

Some antecedents from physical preparation experiences based on the idea of collective sports as complex dynamic systems were presented in the study by Tous-Fajardo et al. (2016) [16]. The authors carried out a comparison of isoinertial vibration platform training (EVT) versus traditional concentric weight training on an under-18 soccer team. The main finding of this study was that the use of isoinertial equipment generates resistances adapted to the individual strength of each participant in the execution of each of the repetitions, so that the overload was dependent on the specific action of the subject and not of the quantitative result of a previous evaluation. The results showed that the EVT group developed better levels in linear displacement as well as in the actions of change of direction and reactive jump. Along with this, their sessions took less time and saved transfer sessions, as the EVT training was functional to the actions of soccer. It is highlighted in this study that the repetition load and intensity were applied within a self-regulation parameter. This was that each player had to perform between six to ten repetitions that allowed them to maintain high power levels, and they were not given a specific amount to execute as during traditional resistance training. These results coincide with those obtained in our study, which fulfills a similar condition regarding the self-regulation of the load intensity, which would suggest that the significant improvements that occurred in power respond effectively to the training program with self-controlled loads.

The fundamental contribution of the present research is the empirical evidence that suggests that the use of the designed program seems to allow that, regardless of the characteristics and previous experiences in strength training of the players, a substantial improvement is achieved both with respect to the increase in power and speed and in the reduction of the loss of these values as the actions are repeated, allowing the players to maintain high levels of these parameters over time, which is a determining factor for performance in a match [34].

One of the main limitations of our research has been the difficulty in isolating the players from their usual handball training, to determine the effect of the training program without other factors that could have conditioned the results. However, as discussed above, the study of Hammami et al. (2018) showed that regular sports training does not affect the results of strength training in a team sport. It has also not been possible to absolutely ensure that the players maintained a regular diet and abstained from consuming stimulant beverages, the consumption of which by university students occurs regularly due to the high load of studies that they do during nighttime [35]. In this context, studies have demonstrated that a high dose of caffeine produces dehydration and reduction of perception of fatigue [36] and that dietary modifications such as a change from a regular diet to one high in fat and the reduction in intake of carbohydrates can influence performance [37], as can other ergogenic aids such nitrate which improves peak power [38]. Any non-reported food or drink that the player may have had before the evaluations, could have affected the results of this study either positively or negatively. These factors will be controlled in future research.

## 5. Conclusions

A program based and designed from the concept of humans as complex dynamic systems, built specifically for the improvement of power, which is controlled under the perception of the effort of each player in an individual and subjective manner, can be an effective instrument to improve the power in counter-movement jumps and the speed of 25 m sprints in university handball players, regardless of their initial physical level and their experience in strength training. In the same way, a program of these characteristics enables a reduction in the loss of jump height and sprint speed after carrying out repetitive actions of these for long times, generating resistance to the repetitive execution of these actions. The main results of this intervention study support further investigations in this line of research focused on self-perceived efforts in strength training for non-professional female handball players in different contexts. The comparison of experimental groups with control groups, and larger sample sizes would provide stronger data and could provide further detail of the specific effects of this approach.

In summary, it is concluded that a program of strength training controlled by self-perceived effort using the OMNI-RES scale, can be a useful instrument for strength and conditioning trainers and handball coaches to improve the repeated shuttle sprint and jump ability of female university handball players. 

## Figures and Tables

**Figure 1 ijerph-18-12662-f001:**
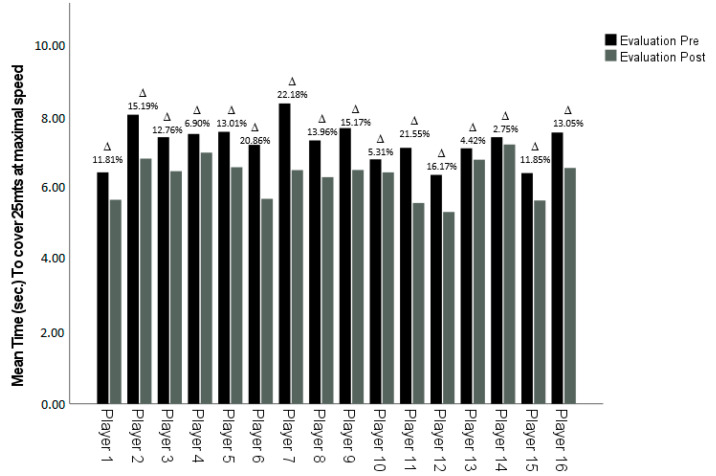
Comparison of pre- versus post- self-regulated load strength training program of the mean time to cover 25 m for each player during the RSSJA test at maximal sprint velocity. Delta (Δ): Percentage of change between pre- and post-evaluation.

**Figure 2 ijerph-18-12662-f002:**
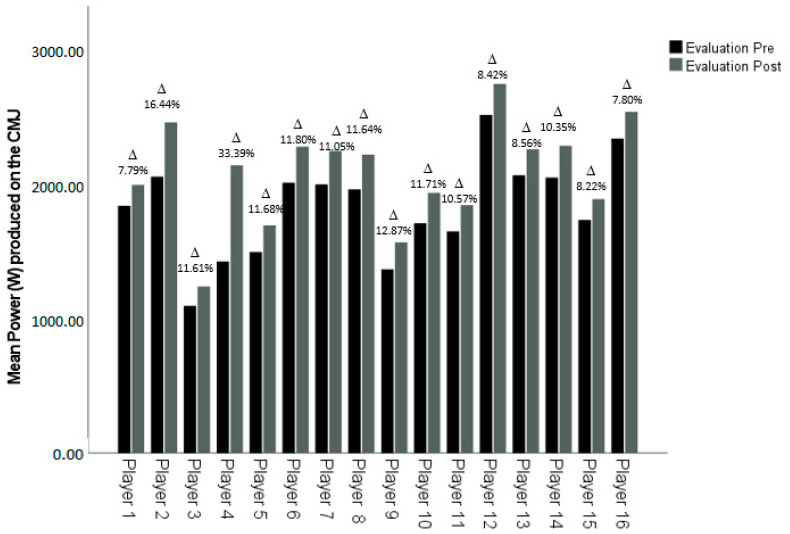
Comparison of pre- versus post- self-regulated load strength training program of the mean power obtained on the counter movement jump of each player during the RSSJA test. Delta (Δ): Percentage of change between pre- and post-evaluation.

**Table 1 ijerph-18-12662-t001:** Training program with level 6–7 OMNI-RES scale.

Goal	Week 1 and 2	Week 3 and 4	Week 5 and 6	Week 7 and 8	Series	Repetitions	Rest
Power	Pull over	Stand pulley Pull over	Disc and elastic band pull over	Stand pull over with unilateral barbells	4	4 to 6	3 to 5 min
Power	Squat	Lunges	Loaded step with bar	Loaded lateral step with full extension	4	4 to 6	3 to 5 min
Power	Chest Press	Inclined chest press	Inclined chest press with barbells	Frontal chest press	4	4 to 6	3 to 5 min
Stabilizer	Frontal plank	Frontal plank	Frontal plank	Frontal plank	1	Until failure	0
Stabilizer	Lateral planks	Lateral planks	Lateral planks	Lateral planks	3	Until failure	0
Compensatory	Biceps curl with barbells	Biceps curl with barbells	Biceps curl with barbells	Biceps curl with barbells	3	12 to 15	30 s to 1 min
Compensatory	Hamstrings	Hamstrings	Hamstrings	Hamstrings	3	12 to 15	30 s to 1 min.
Compensatory	Pull ups (TRX)	Eccentric pull ups (Bar)	Eccentric pull ups (Bar)	Eccentric pull ups (Bar)	3	6 to 8	30 s to 1 min

Descriptive summary of the exercises and their respective loads of the strength training plan.

**Table 2 ijerph-18-12662-t002:** Average pre- and post-test differences for sprints at maximum speed (SMS) and for Counter Movement Jumping (CMJ) of the RSSJA test.

RSSJA	Pre- (n:16)	Post- (n:16)	Difference	*p* Value	CI 95%
SMS 1	6.68 ± 0.61	5.98 ± 0.57	−0.69 ± 0.25	* < 0.001	−0.83/−0.56
SMS 2	6.86 ± 0.56	6.08 ± 0.56	−0.78 ± 0.38	* < 0.001	−0.98/−0.7
SMS 3	7.47 ± 0.79	6.23 ± 0.76	−1.24 ± 0.69	* < 0.001	−1.61/−0.86
SMS 4	7.28 ± 0.66	6.31 ± 0.51	−0.97 ± 0.56	* < 0.001	−1.27/−0.67
SMS 5	7.51 ± 0.59	6.53 ± 0.60	−0.98 ± 0.63	* < 0.001	−1.31/−0.64
SMS 6	7.65 ± 0.61	6.66 ± 0.58	−0.99 ± 0.63	* < 0.001	−1.33/−0.65
CMJ 1	1893.63 ± 335.16	2135.14 ± 349.26	241.51 ± 144.62	* < 0.001	164.44/318.57
CMJ 2	1923.45 ± 401.74	2130.93 ± 402.83	207.48 ± 119.35	* < 0.001	143.88/271.07
CMJ 3	1872.88 ± 348.51	2097.81 ± 384.16	224.93 ± 146.97	* < 0.001	146.61/303.25
CMJ 4	1814.04 ± 387.16	2050.31 ± 363.83	236.27 ± 171.08	* < 0.001	145.10/327.44
CMJ 5	1775.11 ± 410.27	2073.23 ± 417.47	298.11 ± 188.82	* < 0.001	197.49/398.73
CMJ 6	1739.72 ± 396.59	2045.99 ± 409.79	306.27 ± 201.28	* < 0.001	199.01/413.52

Data is presented as mean and SD; *: Significant value for *p* < 0.05 of the T Student test for related samples. SMS = sprint at maximal speed; CMJ = counter movement Jump. SMS’s unit of measure is presented as seconds. CMJ’s unit of measure is presented as Watts.

**Table 3 ijerph-18-12662-t003:** The effect size of each player for the counter movement jump and sprint at maximal speed.

CMJ					SMS			
Subjects	Stand.	Cohen’s d	Confidence Interval.		Stand.	Cohen’s d	Confidence Interval.	
			Lower	Upper			Lower	Upper
Player 1	52.32386	−2.976	−4.657	−1.232	0.36841	2.058	0.588	3.467
Player 2	68.52510	−5.916	−8.685	−3.103	0.59879	2.035	0.570	3.437
Player 3	138.79780	−1.039	−2.235	0.200	0.36205	2.606	0.977	4.170
Player 4	117.30827	−6.106	−8.951	−3.219	0.68133	0.758	−0.493	1.947
Player 5	95.87270	−2.068	−3.479	−0.595	0.34579	2.839	1.139	4.476
Player 6	69.46183	−3.880	−5.872	−1.830	0.56631	2.646	1.005	4.222
Player 7	145.35929	−1.712	−3.035	−0.330	0.49080	3.766	1.756	5.718
Player 8	131.90675	−1.963	−3.347	−0.518	0.50538	2.018	0.558	3.416
Player 9	114.65234	−1.764	−3.099	−0.370	0.31462	3.687	1.705	5.610
Player 10	116.08309	−1.958	−3.341	−0.514	0.45312	0.794	−0.406	1.959
Player 11	100.11270	−1.952	−3.333	−0.510	0.55133	2.775	1.095	4.392
Player 12	97.60512	−2.374	−3.869	−0.815	0.30314	3.387	1.507	5.206
Player 13	43.12921	−4.494	−6.713	−2.222	0.28755	1.090	−0.159	2.293
Player 14	66.00375	−3.593	−5.483	−1.643	0.50826	0.400	−0.754	1.535
Player 15	52.32386	−2.976	−4.657	−1.232	0.36841	2.058	0.588	3.467
Player 16	1.57945	−2.068	−3.479	−0.595	0.34579	2.839	1.139	4.476

Note: Cohen’s d = large effect >0.80; medium effect 0.50 < 0.80; small effect 0.20 < 0.50. Cohen’s d uses combined standard deviation.

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
