# Peer review of "Effects of a Self-Regulated Training Program on the Repeated Power in Female College Handball Players: An Intervention Study"

_ijerph, 2021, doi:10.3390/ijerph182312662_

Round 1

Reviewer 1 Report

The topic is interesting to readers.
This paper may reach a publishable level only with substantial revisions.
My comments / questions:
If the authors provide data from a pilot study, what is the purpose of this study? What are the planned studies in the future ? I didn't find the answer to this question in the article provided.
In the discussion, the author mentions the dietary factor of the subjects, but it is unclear what effect the dietary factor had on the reliability of the study results.
The author uses the terms physical properties but does not provide the concepts of these terms (Speed and power, power and strange). When analyzing the research data, there are blurred boundaries between the traits being developed. Relevant sources must be considered in this regard. For example, Bompa, T. O., & Buzzichelli, C. (2019). Periodization-: theory and methodology of training. Human kinetics.
Terminological clarity is needed.
The author does not provide the exclusivity of the proposed training. The conclusions should explain how this study contributes to the training of athletes and ask questions for future research.
Sincerely

Author Response

Dear reviwer,

Dear reviewer,

Would like to express our gratitude for the time taken to review our manuscript and for the comments made, which we believe to be critical for producing rigorous and quality research. We have detailed below the changes made in the original article: Effects of a Self-Regulated Training program on the repeated Power in Collegiate Female Handball Players: An Intervention Study (ijerph-1444342).

Modifications have been made in the original manuscript following the three reviewers’ comments. For each modification we have written: the original comment as written by the reviewer in addition to the line number; and the change made in response to that comment.

MODIFICATIONS

 We will assume all comments from reviewers in order to improve the quality of content, differentiator elements, and guidance for future research.

Regarding your comments, modifications are explained detailed next:

Comment 1:

If the authors provide data from a pilot study, what is the purpose of this study? What are the planned studies in the future? I didn't find the answer to this question in the article provided.

Response 1:

As reviewer 2 has indicates that this is not a pilot study but an intervention study one, we have made this change in the title as well in the study design in line 95.

The purpose of our study is now “to describe the effects of a resistance training program whose loads are controlled by self-perceived effort of a group of Chilean university female handball players, on their repeated shuttle sprint and jump ability. And also, with the main results, determinate the continuity of this line of investigation focused on self-perceived efforts to control strength training loads in female handball players of different non-professional contexts”. As you can see in line 86 to 92 and then is highlighted again in line 282 to 284.

Comment 2:

In the discussion, the author mentions the dietary factor of the subjects, but it is unclear what effect the dietary factor had on the reliability of the study results.

Response 2:

The effect of the dietary factors are now mentioned, described and supported by a cited bibliography: “…In this context, studies have demonstrated that high dose of caffeine produces dehydration and reduction of perception of fatigue [35], and that dietary modifications such change from regular diet to highly fat consumption and reduction of carbohydrates influence performance [36], as well as other ergogenic aids such nitrate which improves peak power [37]. Any non-reported food or drink that the player may have had before the evaluations, could have affected both positively or negatively the results of this study, which will be controlled on future research”. You can find this between lines 365 and 371.

Comment 3:

The author uses the terms physical properties but does not provide the concepts of these terms (Speed and power, power and strange). When analyzing the research data, there are blurred boundaries between the traits being developed. Relevant sources must be considered in this regard. For example, Bompa, T. O., & Buzzichelli, C. (2019). Periodization-: theory and methodology of training. Human kinetics.
Terminological clarity is needed.

Response 3:

 The referred concepts are now defined and supported by relevant bibliography. This between lines 182 and 189.

Comment 4:

The author does not provide the exclusivity of the proposed training. The conclusions should explain how this study contributes to the training of athletes and ask questions for future research.

Response 4:

The study contribution is now explained: “… As this is an intervention study, main results allow us to determinate the continuity of the line of research focused on self-deceived efforts on strength training for non-professional female handball players on different context. This, in order to provide strong data of the specific effects by comparing experimental groups with control groups.

Finally, it is concluded that a program of strength training controlled by self-perceived effort using the OMNI-RES scale, can be a useful instrument for strength & condinioning trainers and handball coaches to improve the repeated shuttle sprint and jump ability of female university handball players”. This can be found between lines 392 and 395.

Reviewer 2 Report

Congratulations by your important research and great contribution for handball and coletive sports. Considering the high quality of your manuscript, my comments are brief. Thank you very much. It is beautiful work and enjoyable/directly reading. The suggestions are below:

 Line 90: It is not a pilot study (you cannot reduce the importance of your research), it’s an intervention study. I think that this study is a quasi-experimental research with a specific population, a unique group, and it is not a sample (athletes from a team can be considered a singular population because there is not another group similar to them).

Line 175: I would like to suggest in the sense to improve the analysis quality of data. The authors should include a real measure of intervention effect power, like Cohen’s D and individual delta % effect for each player (a bar in the graph to each one of 16 players with delta and pre to post-test result). Only the p-value don’t represent the effect of an intervention because it is a measure most linked with sample size error (type I or II) than effect size. A interesting read about this kind of methodology of results representation: https://www.frontiersin.org/articles/10.3389/fphys.2018.01443/full

It is aligned to your study perspective preserving the theoretical approach of self-regulation and biological individuality.

Line 300: I suggest change: This pilot study reveals positive data that impulses to explore this line through ex(…) by: This intervention study reveals positive data that impulses to explore this line through ex (…)

Line 304: conclusions could be most directly according your objectives and contributions to teachers and coaches of handball.

Author Response

Dear reviewer,

Would like to express our gratitude for the time taken to review our manuscript and for all the detail comments you have made, which we are sure that they will be critical for producing rigorous and quality research. We have detailed below the changes made in the original article: Effects of a Self-Regulated Training program on the repeated Power in Collegiate Female Handball Players: A Pilot Study (ijerph-1444342).

Modifications have been made in the original manuscript following the three reviewers’ comments. For each modification we have written: the original comment as written by the reviewer in addition to the line number; and the change made in response to that comment.

MODIFICATIONS

We will assume all comments from reviewers in order to improve the quality of content, differentiator elements, and guidance for future research.

Regarding your comments, modifications are explained detailed next:

Comment 1:

Line 90: It is not a pilot study (you cannot reduce the importance of your research), it’s an intervention study. I think that this study is quasi-experimental research with a specific population, a unique group, and it is not a sample (athletes from a team can be considered a singular population because there is not another group similar to them).

Response 1:

We have changed the title of our research and now is “Effects of a Self-Regulated Training program on the repeated Power in Collegiate Female Handball Players: An intervention Study

We have also made modification in the material and method section where now we refer our study as an intervention one. You can see this modification in the line 95 and in line 372.

Comment 2:

Line 175: I would like to suggest in the sense to improve the analysis quality of data. The authors should include a real measure of intervention effect power, like Cohen’s D and individual delta % effect for each player (a bar in the graph to each one of 16 players with delta and pre to post-test result). Only the p-value don’t represent the effect of an intervention because it is a measure most linked with sample size error (type I or II) than effect size. A interesting read about this kind of methodology of results representation: https://www.frontiersin.org/articles/10.3389/fphys.2018.01443/full

It is aligned to your study perspective preserving the theoretical approach of self-regulation and biological individuality.

Response 2:

We have added your indications and now is possible to see the date obtained by Cohen’s d and the delta values for each player in the table 3 and figures 1 & 2. You can check this between lines 214 and 278

Comment 3:

Line 300: I suggest change: This pilot study reveals positive data that impulses to explore this line through ex(…) by: This intervention study reveals positive data that impulses to explore this line through ex (…)

Response 3:

We have made the change you suggested and now it says “…intervention study…”. You can see this modification in line 372

Comment 4:

Line 304: conclusions could be most directly according to your objectives and contributions to teachers and coaches of handball.

Response 4:

We have added another paragraph to the conclusion where it says: “Finally, it is concluded that a program of strength training controlled by self-perceived effort using the OMNI-RES scale, can be a useful instrument for strength & conditioning trainers and handball coaches to improve the repeated shuttle sprint and jump ability of female university handball players.” This modification can be found between lines 392 and 395.

Reviewer 3 Report

Thank you for the work you have done on this paper, which reports the results of an interesting training program, with potential for broader application. As such, the paper has the potential to make a contribution to the literature. At this stage, however, more work is required.  In particular, I would ask you to consider the following points.

  • The main challenge for the paper at present relates to the limitations you have identified in lines 293-299, in particular the recognition that it was not possible to isolate the players from their usual training. This limitation, as currently discussed, means we are not able to say whether the changes observed were due to the intervention or were due to the regular training, or some combination of the two. It is important that you discuss this and justify this some more, both in the discussion of the study design and in this final section.  
  • Line 77 reports that there is a paradigm shift. In the discussion prior to this statement, more clarity is required over (1) what this paradigm shift is; and (2) the evidence that such a shift exists.
  • The balance between results and discussion needs some work. In the results section, more discussion is needed of what the table and figures are showing. This can be balanced by reducing some of the discussion of other papers in the discussion section.
  • Line 281-286: this paragraph sets out the fundamental contribution of the paper. On reading this paragraph, however, it is not clear what is the contribution. The paragraph needs work to clarify its meaning.
  • Overall, attention is needed to the written expression. For example: line 65, should be 'isoinertial'; line 67: ‘an’ should be ‘a’; line 69: add ‘or herself’ after ‘himself’; line 90: ‘wich’ should be ‘which’. These are just some examples. The whole paper requires careful proof reading
  • In Part 5, Conclusions, please provide more specific detail on how the study connects to the paradigm shift identified earlier in the paper.

Author Response

Dear reviewer,

Would like to express our gratitude for the time taken to review our manuscript and for all the detail comments you have made, which we are sure that they will be critical for producing rigorous and quality research. We have detailed below the changes made in the original article: Effects of a Self-Regulated Training program on the repeated Power in Collegiate Female Handball Players: A Pilot Study (ijerph-1444342).

Modifications have been made in the original manuscript following the three reviewers’ comments. For each modification we have written: the original comment as written by the reviewer in addition to the line number; and the change made in response to that comment.

MODIFICATIONS

We will assume all comments from reviewers in order to improve the quality of content, differentiator elements, and guidance for future research.

Regarding your comments, modifications are explained detailed next:

Comment 1:

The main challenge for the paper at present relates to the limitations you have identified in lines 293-299, in particular the recognition that it was not possible to isolate the players from their usual training. This limitation, as currently discussed, means we are not able to say whether the changes observed were due to the intervention or were due to the regular training, or some combination of the two. It is important that you discuss this and justify this some more, both in the discussion of the study design and in this final section.

Response 1:

A similar comment was made for the first reviewer who have suggested that we should explain and support what we have declared about our study limitations. For this, we have added the next paragraph:

“However, as discussed above, the study of Hammami et al., (2018) has shown that regular sport training does not affect the results of strength training in team sport. It has also not been possible to firmly ensure that the players maintained a regular diet and that, although they have been asked not to consume stimulant beverages, their consumption by university students occurs regularly due to the high load of studies that they do during nighttime [34]. In this context, studies have demonstrated that high dose of caffeine produces dehydration and reduction of perception of fatigue [35], and that dietary modifications such change from regular diet to highly fat consumption and reduction of carbohydrates influence performance [36], as well as other ergogenic aids such nitrate which improves peak power [37]. Any non-reported food or drink that the player may have had before the evaluations, could have affected both positively or negatively the results of this study, which will be controlled on future research.”

These changes can be seen between lines 361 and 371.

Comment 2:

Line 77 reports that there is a paradigm shift. In the discussion prior to this statement, more clarity is required over (1) what this paradigm shift is; and (2) the evidence that such a shift exists.

Response 2:

We have noted that this is not really a change of paradigm but an alternative way to plan strength training in team sport. In this sense, we have changed the paragraph to the next:

 “This evidence provides empirical arguments about the usefulness of this alternative way to perform strength training in team sports”

This can be seen in lines 78 to 79

Comment 3:

The balance between results and discussion needs some work. In the results section, more discussion is needed of what the table and figures are showing. This can be balanced by reducing some of the discussion of other papers in the discussion section.

Response 3:

The second reviewer have suggested some changes in the results presentation, where now data shows the effect size of the experiment on each player. These results were obtained by using Cohen’s d and delta value. In this sense, more discussion has been made.  This can be found between lines 192 and 278

Comments 4:

Line 281-286: this paragraph sets out the fundamental contribution of the paper. On reading this paragraph, however, it is not clear what is the contribution. The paragraph needs work to clarify its meaning.

Response 4:

To clarify the fundamental contribution of the paper, a new paragraph was added between lines 348 and 354 which says the next:

“The fundamental contribution of the present research is the empirical evidence that suggests that the use of the designed program seems to allow that, regardless of the characteristics and previous experiences in strength training that the players have, a substantial improvement is achieved both with respect to the increase in power and speed and in the reduction of the loss of these values as the actions are repeated, allowing to maintain high levels of these parameters over time, which is a determining factor for performance in a match [33].”

Comment 5:

Overall, attention is needed to the written expression. For example: line 65, should be 'isoinertial'; line 67: ‘an’ should be ‘a’; line 69: add ‘or herself’ after ‘himself’; line 90: ‘wich’ should be ‘which’. These are just some examples. The whole paper requires careful proof reading

Response 5:

Careful proof reding was made after your comments, several changes were made.

Comment 6:

In Part 5, Conclusions, please provide more specific detail on how the study connects to the paradigm shift identified earlier in the paper.

Response 6:

As we have changed the concept and now there is no change of paradigm. It seems not necessary to clarify this point.

Round 2

Reviewer 1 Report

Thanks for replying to my comments.
I did not understand your 3rd comment.
Please provide definitions of the training characteristics of your proposed methods according to the criterion of physical intensity. The descriptions provided lack clarity.

You can use the example:
Rapoport, B. I. (2010). Metabolic factors limiting performance in marathon runners. PLoS computational biology, 6 (10), e1000960.
Kravitz, L. Biological Energy Used for Anaerobic Training.

Your manuscript does not provide enough information on what physical characteristics are sought to be trained.
Please provide definitions: what do you mean?
1. Speed and power,
2. Power and strength
by the terms of metabolic parameters of energy production and authors.

Sincerely. 

Author Response

Dear reviewer,

Would like to express again our gratitude for the time taken to review our manuscript and for the comments made, which we believe to be critical for producing rigorous and quality research. We have detailed below the changes made in the original article: Effects of a Self-Regulated Training program on the repeated Power in Female College Handball Players: An Intervention Study (ijerph-1444342).

Modifications have been made in the original manuscript following the three reviewers’ comments. For each modification we have written: the original comment as written by the reviewer in addition to the line number; and the change made in response to that comment.

MODIFICATIONS

Regarding your comments in which you have said:

Please provide definitions of the training characteristics of your proposed methods according to the criterion of physical intensity. The descriptions provided lack clarity.

You can use the example:
Rapoport, B. I. (2010). Metabolic factors limiting performance in marathon runners. PLoS computational biology, 6 (10), e1000960.
Kravitz, L. Biological Energy Used for Anaerobic Training.

Your manuscript does not provide enough information on what physical characteristics are sought to be trained.
Please provide definitions: what do you mean?
1. Speed and power,
2. Power and strength
by the terms of metabolic parameters of energy production and authors.

Response:

In a deeper reading, we have noticed that we were using 2 different concepts when we were referring about the kind of training we were using. We have now unified these concept as “strength training”.

Also, we have added a new section between lines 166 y 178, that explains the relationship between power, speed and strength.

Looking forward for any new comment you have.

Kind regards

Reviewer 3 Report

The authors have addressed the comments appropriately, thank you.  A very close final proof reading is required.

Author Response

Dear Reviwer,

We would like to thank you once again for your review.

As you requested, we have asked to a english native speaker to proof read our article.

Major changes where made.

Kind regards